# Enhanced Private Sector Engagement for Tuberculosis Diagnosis and Reporting through an Intermediary Agency in Ho Chi Minh City, Viet Nam

**DOI:** 10.3390/tropicalmed5030143

**Published:** 2020-09-14

**Authors:** Luan Nguyen Quang Vo, Andrew James Codlin, Huy Ba Huynh, Thuy Doan To Mai, Rachel Jeanette Forse, Vinh Van Truong, Ha Minh Thi Dang, Bang Duc Nguyen, Lan Huu Nguyen, Tuan Dinh Nguyen, Hoa Binh Nguyen, Nhung Viet Nguyen, Maxine Caws, Knut Lonnroth, Jacob Creswell

**Affiliations:** 1Friends for International TB Relief, Ha Noi 100000, Vietnam; andrew.codlin@tbhelp.org (A.J.C.); huy.huynh@tbhelp.org (H.B.H.); thuy.mai@tbhelp.org (T.D.T.M.); rachel.forse@tbhelp.org (R.J.F.); 2Interactive Research and Development, Singapore 189677, Singapore; 3Pham Ngoc Thach Hospital, Ho Chi Minh City 700000, Vietnam; drvinhpnt2020@gmail.com (V.V.T.); hadtm2018@gmail.com (H.M.T.D.); mmdbang@gmail.com (B.D.N.); nguyenhuulan1965@gmail.com (L.H.N.); 4Viet Nam National Lung Hospital, Ha Noi 100000, Vietnam; tuandinh10@yahoo.com (T.D.N.); nguyenbinhhoatb@yahoo.com (H.B.N.); vietnhung@yahoo.com (N.V.N.); 5Liverpool School of Tropical Medicine, Department of Clinical Sciences, Liverpool L3 5QA, UK; maxine.caws@lstmed.ac.uk; 6BNMT Nepal, Lazimpat, Ward No. 2, Kathmandu 44600, Nepal; 7Karolinska Institutet, Department of Public Health Sciences, 171 77 Solna, Sweden; knut.lonnroth@ki.se; 8Stop TB Partnership, 1218 Geneva, Switzerland; jacobc@stoptb.org

**Keywords:** tuberculosis, private sector, intermediary agency, referral, notification, Viet Nam

## Abstract

Under-detection and -reporting in the private sector constitute a major barrier in Viet Nam’s fight to end tuberculosis (TB). Effective private-sector engagement requires innovative approaches. We established an intermediary agency that incentivized private providers in two districts of Ho Chi Minh City to refer persons with presumptive TB and share data of unreported TB treatment from July 2017 to March 2019. We subsidized chest x-ray screening and Xpert MTB/RIF testing, and supported test logistics, recording, and reporting. Among 393 participating private providers, 32.1% (126/393) referred at least one symptomatic person, and 3.6% (14/393) reported TB patients treated in their practice. In total, the study identified 1203 people with TB through private provider engagement. Of these, 7.6% (91/1203) were referred for treatment in government facilities. The referrals led to a post-intervention increase of +8.5% in All Forms TB notifications in the intervention districts. The remaining 92.4% (1112/1203) of identified people with TB elected private-sector treatment and were not notified to the NTP. Had this private TB treatment been included in official notifications, the increase in All Forms TB notifications would have been +68.3%. Our evaluation showed that an intermediary agency model can potentially engage private providers in Viet Nam to notify many people with TB who are not being captured by the current system. This could have a substantial impact on transparency into disease burden and contribute significantly to the progress towards ending TB.

## 1. Introduction

Tuberculosis (TB) is a curable disease, yet an estimated 10 million people develop active TB and 1.5 million people succumb to TB each year [1]. It remains the deadliest disease caused by a single infectious agent and a major source of avoidable deaths worldwide. Over the first 6 months of 2020, an estimated 867,000 persons died of TB as a negative consequence of the Covid-19 pandemic [2]. Moreover, it is estimated that about one-quarter of the world’s population is infected with subclinical, noninfectious TB [3].

Viet Nam has a well-organized National Tuberculosis Control Programme (NTP). TB treatment is provided free of charge at all public sector sites and reported TB treatment outcomes are high. In 2014, Viet Nam committed to reduce TB prevalence to 20 per 100,000 by 2030 [4]. However, following the second national prevalence survey in 2018 [5], the country’s estimated TB incidence rate was revised upward from 124 to 174 per 100,000, suggesting that just 57% of the estimated burden was captured by the NTP’s official TB case notification statistics [6].

In 1986, the government initiated a package of reforms [7] that shifted Viet Nam’s centralized, public healthcare sector towards neoliberalism and later to New Public Management, which resulted in the rapid development of a private healthcare sector [8]. Fueled by the country’s strong economic growth and rising population welfare, demand for private sector services due to greater convenience and quality perceptions has similarly increased [9].

People often prefer to seek care with non-NTP facilities owing to their flexibility regarding diagnostic procedures, drug regimens and treatment observation methods, more convenient operating hours and locations, and lower administrative burden [10]. Studies have shown that pharmacies and private clinics represent the initial point of health care seeking for 50−70% of people with TB in Viet Nam [11,12,13]. Despite the existence of a mandatory notification law since 2007 [14], the implementation of this policy has been suboptimal. As such, the private healthcare sector is a major driver of ‘missed people with TB’ and loss to follow-up (LTFU) [15].

The Ministry of Health subsequently passed a law in 2013 (Circular 02/2013/TT-BYT) to enable systematic inclusion of private providers and public institutions outside of the NTP via four public-private mix (PPM) engagement models: (1) referral; (2) diagnosis; and referral; (3) directly observed treatment (DOT) provider and (4) full-service TB care facility [16]. The law has resulted in roughly 10% of TB case notifications at the national-level coming from PPM initiatives. Yet over 80% of these PPM notifications originate from public institutions, such as general care, military, and police hospitals that are not specialized in TB care and thereby are outside of the technical supervision of the NTP. This implies that private providers contribute only 2% to annual notifications nationwide. Meanwhile, it is estimated that about half of Viet Nam’s ‘missing cases’ are taking their TB treatment outside of the NTP network [17]. More importantly, there is evidence that private sector TB care is often of substandard quality. Patients may suffer diagnostic delays with no bacteriological confirmation and receive inappropriate or inadequate treatment regimens. Poor adherence support has resulted in loss to follow-up rates of up to 65% [18,19,20,21].

A key reason for the limited engagement of private providers is the restrictive nature of Circular 02/2013/TT-BYT. Providers that wish to retain their clientele are expected to participate as a full-service TB facility and fulfill associated diagnostic and reporting requirements, while submitting to close oversight and supervision by the NTP. Meanwhile, benefits of participation, such as capacity building, free medicines, and eligibility for monetary stipends at government rates, may be insufficiently powered or implemented [16]. This has proven untenable for many non-NTP providers apart from large public tertiary care facilities. As a result, it is critical to develop and evaluate engagement schemes, which take into account the economic interests of smaller private providers.

One such scheme is the Private Provider Interface Agency (PPIA) model that has subsequently been scaled through the Joint Effort for Elimination of Tuberculosis (JEET) to 23 states of India [22,23]. This model employs intermediary agencies [15,24] that aim to offer a tangible value proposition with bottom-line impact rather than appeal to altruistic motivations [25,26]. This value proposition includes free or discounted access to nucleic acid amplification testing (NAAT) and medicines at pre-negotiated price-points for providers and patients and, perhaps most importantly, the option for private providers to retain their customers and thereby their livelihoods [27]. The implementation of PPIA’s showed promising results in multiple sites throughout India [28,29] and has been recognized as one avenue of sustainably scaling private sector engagement for TB worldwide [30].

In 2017, Friends for International TB Relief piloted a private-sector engagement initiative called Proper Care Private Sector (PCPS), modeled after the successful PPIA pilots from India [27]. This pilot investigated the feasibility of building a portfolio of private providers and measured the outputs of incentivizing and supporting referral and reporting of private TB treatment.

## 2. Materials and Methods

### 2.1. Study Setting

This pilot was conducted in two districts of Ho Chi Minh City (HCMC), Viet Nam—District 10 and Go Vap—between July 2017 and March 2019. The intervention area had a combined population of 1.2 million people and notified 1070 people with All Forms of TB in the 12 months preceding the study. In each district, there is a District TB Unit (DTU) responsible for managing diagnosis, treatment and notification of TB according to NTP guidelines and for coordinating patient management with primary health facilities. There were no official private sector TB-reporting entities in the evaluation area before this study’s implementation.

### 2.2. Private Provider Engagement

We obtained lists of licensed private healthcare providers from each intervention district’s regulatory authority. These providers included pharmacies, single-doctor practices and multi-doctor clinics. In collaboration with licensing, health, and TB authorities, through consensus we conducted a mapping exercise to identify priority providers with a high likelihood of encountering people who had pulmonary TB, while categorically excluding certain specialists, such as dermatologists, obstetricians, and gynecologists. Through repeated in-person and telephonic engagement, we recruited eligible providers. Interested providers were invited to capacity building events organized in collaboration with the Pham Ngoc Thach provincial lung hospital (PNT). The scope of these training events included new diagnostic tests for TB and specifically Xpert MTB/RIF (Xpert) and the newly recommended MTB/RIF Ultra assay [31], standardized TB treatment regimens, and follow-up schedules according to NTP guidelines. We complemented these formal training events with one-on-one provider detailing activities [32] to elaborate on the study’s procedures, the provider’s role and responsibilities, and the benefits of participation. Providers were eligible to participate through two principal strategies: diagnostic referral and private TB treatment reporting (Figure 1).

### 2.3. Diagnostic Referral Strategy

In this strategy, participating providers verbally screened their customers for TB symptoms and distributed referral vouchers to anyone reporting at least one TB symptom, i.e., (productive) cough or hemoptysis, weight/appetite loss, fatigue, fever, night sweats, chest pain, dyspnea. Symptomatic persons could use the voucher to access a chest X-ray (CXR) subsidy of VND 50,000 (USD 2.20 at an exchange rate of VND 22,700 = USD 1) at one of the study’s 12 participating radiology sites. As the cost per CXR charged by these radiology sites ranged from VND 80,000 (USD 3.52) to VND 120,000 (USD 5.29), the radiography site collected the balance payment from the health-seeking person. In comparison, the price for one CXR at the District TB Unit was VND 49,000 (USD 2.16) at the start of the study and was subsequently raised to VND 69,000 (USD 3.04). Patients who elected to take their TB treatment with a private provider were charged a consultation fee of between VND 80,000 and VND 150,000 (USD 6.61) in addition to drugs and other services. According to field staff estimates, the approximate average cost per visit per person at private facilities was VND 200,000 (USD 8.81).

Persons assessed with parenchymal abnormalities on CXR by the X-ray technician and verified by the attending radiologist at the radiography site provided a sputum sample for free follow-on testing with the Xpert assay. At selected sites, health-seeking persons also underwent smear microscopy, in which case these results were requested from the participating provider as well. Sputum was collected at the radiography site or by the referring private provider. Study staff collected sputum specimens for transport to a designated government Xpert laboratory in Go Vap district. People with Xpert-positive results were encouraged to take treatment at their closest DTU, or at PNT if their Xpert result showed rifampicin resistance. When an individual was diagnosed and treated for TB via this strategy, the private provider making the initial referral received a VND 500,000 (USD 22.07) payment or approximately 2.5x the estimated average cost per visit per person. If the person chose to take TB treatment with a private provider, the treatment was recorded through the study’s second strategy.

### 2.4. Private TB Treatment Reporting Strategy

The second strategy focused on documenting private TB treatment practices. Once a month, study staff collected TB treatment information from participating private providers. This information included individuals diagnosed through the diagnostic referral strategy above that elected treatment outside of the NTP. Providers were paid VND 500,000 (USD 22.07) for each complete patient report, which included the patient’s name, age, sex, address, CXR results, sputum test results (Xpert, smear, culture, other), type of TB (pulmonary, extra-pulmonary), treatment regimen, and initiation dates. Treatment outcomes were not systematically assessed in this pilot study due to resource limitations and data provided by providers were sparse as providers did not conduct post-treatment follow-up with patients.

Despite the attempts to characterize these treatment reports in detail, they were not recognized by the NTP for official notification for several reasons. The primary reason was that these providers were not registered as official PPM model 4 participants in accordance to 02/2013/TT-BYT and therefore had not undergone required capacity building and site assessment by the NTP.

### 2.5. Statistical Analyses

We tabulated descriptive statistics for private provider engagement and participation, the number and proportion of referred people progressing through the study’s TB care cascade by intervention district and the private TB treatment reported to our study. We calculated the ratio of bacteriologic confirmation over the number of successful CXR referrals. Official TB notifications were collected from the two intervention districts for three years prior to the study and during the study period to analyze trends of official TB notifications before and during the pilot. Additional notifications and percent change from baseline were calculated using a pre-/post-intervention comparison of official notification data in the intervention districts. Due to barriers outlined above, the collected private TB treatment cases were not included in the official NTP notification statistics, so that a second additionality model was constructed to assess the impact of including these privately treated individuals in official TB statistics for the intervention districts. Statistical analyses were performed on Stata version 13 (StataCorp, College Station, TX, USA).

### 2.6. Ethical Considerations

The Institutional Review Boards of Pham Ngoc Thach Hospital (155/NCKH-PNT) and the Hanoi School of Public Health (324/2019/YTCC-HD3) granted scientific and ethical approval for this study. The Ho Chi Minh City Provincial People’s Committee approved the implementation of the intervention (4699/QD-UBND). Participating private providers granted permission to use data for the analyses based on the terms and conditions of their practice. All personally identifying information was removed prior to analysis.

## 3. Results

### 3.1. Private Provider Engagement and Participation

The study enumerated 1107 licensed private providers in the two intervention districts (Table 1). Of these, 67.0% (742/1107) were targeted for recruitment based on the initial mapping exercise and 53.0% (393/742) of those targeted agreed to participate. Among participants, at least one staff member of 48.6% of centers (191/393) attended a capacity building event. By the end of the study, we recorded at least one referral for CXR from 32.1% (126/393).

Of the 126 private providers with at least one successful CXR referral (Table 2), 58.7% were multi-doctor clinics and 25.4% were single-doctor practices. These two provider types accounted for 70.0% and 18.9% of referrals, respectively. The remaining referrals were from pharmacies, hospitals or could not be traced to the source. The bacteriologic positivity rate among successful CXR referrals was highest among single pulmonologist practices at 58.6%, followed by multi-doctor clinics at 21.9% and single-doctor practices with no specialty focus at 11.2%. Eighty-two point two percent of the people diagnosed with TB via the diagnostic referral strategy were referred by just ten private providers constituting 7.9% (10/126) of those making at least one successful CXR referral and 2.5% (10/393) of those signing participation agreements.

The study received TB diagnosis and treatment data from 3.6% (14/393) of participating private providers. These consisted of 71.4% (10/14) single-doctor practices and 28.6% (4/14) multi-doctor clinics. The top five providers supplying TB diagnosis and treatment data reported 81.7% (907/1112) of patients on private TB treatment.

### 3.2. Detection and Reporting Yield

The study identified 1203 people with TB of whom 7.6% (91/1203) were referred and linked to care with the NTP (Figure 2), while 92.4% (1112/1203) consisted of private TB treatment reports and remained un-notified (Table 3). All 91 TB patients linked to care with the NTP were bacteriologically confirmed. Among persons treated in the private sector, the proportion with bacteriologic confirmation was 30.5% (339/1112). Together, the total proportion of TB patients with bacteriologic confirmation was 35.7% (430/1203). Overall, 1.2% (15/1203) were people with Multi-drug resistant TB (MDR-TB). Patients diagnosed with rifampicin resistance were largely referred by private providers to NTP facilities. Particularly, diagnostic referrals generated 93.3% (14/15) of persons detected with rifampicin resistance (Figure 2). Meanwhile, private TB treatment reports included one MDR-TB case (Table 3). In addition to persons treated for active TB, four persons were treated for latent TB infection by private providers.

The results of the study’s diagnostic referral strategy are in Figure 2. The 12 radiology centers recorded 4984 CXR results, of which 817 were abnormal (16.4% of those with CXR results). Sputum specimens were collected from 65.4% (534/817) of these individuals and tested on the Xpert assay with a positivity of 25.8% (138/534) including 14 individuals with rifampicin-resistant TB (14/138 = 10.1%). An additional 528 smear microscopy tests were conducted for individuals who did not get a CXR or presented no radiographic abnormalities suggestive of TB but still reported TB symptoms, resulting in the detection of 31 (31/528 = 5.9%) people with smear-positive TB. Of the total 169 people diagnosed with bacteriologically-confirmed TB, 95.9% (162/169) were linked to care, corresponding to a ratio of 3.2% among successfully referred persons with a CXR screen. Among patients linked to care, 56.2% (91/162) were initiated on treatment at a NTP facility, while 43.8% (71/162) elected to take treatment with the initially referring private provider. These patients are included in the private TB treatment reports.

The characteristics of the privately-treated, un-notified 1112 individuals are in Table 3. Of these, 30.5% (339/1112) had either a positive smear microscopy, Xpert, and/or culture result. Just 29.0% (322/1112) of those taking private TB treatment lived inside the study’s intervention area, with another 41.1% (455/1112) living in one of HCMC’s other 22 districts. About 28.7% (319/1112) of privately treated persons were registered residents of other provinces, while the remaining 1.3% (14/1112) of people had no documented address. Overall, 68.3% (759/1112) of people privately treated for TB were prescribed a standard first-line regimen as per NTP guidelines, while the records for another 27.6% (307/1112) of people showed the correct drugs but were modified from the standard regimen or missing information on duration. Three percent (33/1112) of treatments included streptomycin, and 1.1% (12/1112) included levofloxacin.

### 3.3. Notification Impact

Table 4 and Figure 3 summarize changes in the NTP’s TB case notifications in the study’s intervention area and present the modeled impact of including private TB treatment on official notification statistics. Bacteriologically-confirmed and All Forms TB notifications increased by +17.0% (+177 TB cases) and +8.5% (+158 TB cases), respectively, over six quarters of implementation. If private TB treatment had been eligible for inclusion in the official notification statistics, bacteriologically-confirmed and All Forms of TB notifications would have increased by +49.7% (+516 TB cases) and +68.3% (+1270 TB cases), respectively.

## 4. Discussion

Our pilot study showed that the PPIA model was effective in engaging a large number of private providers in the Vietnamese urban setting to contribute to TB care and prevention efforts. We found a substantial number of persons treated for TB in the private sector of HCMC, the vast majority of whom were not known to the NTP. This indicates that creating enabling mechanisms, as well as further scale-up and evaluation of private TB treatment reporting approaches, should be a critical component of the TB response in Viet Nam’s urban areas.

Numerous studies have shown that effective engagement of private providers to screen for TB and refer presumptive cases for diagnostic testing can be an efficient way to close the detection gap [33,34,35]. This was corroborated by the results of our study and particularly by the increase in All Forms TB notifications compared to the baseline period. Moreover, this share of private provider contribution to notifications (+8.5%) was over five times Viet Nam’s 2017 national average private sector contribution rate (1727/105,733 = 1.6%) [36]. Lastly, and perhaps most telling, un-notified private TB treatment reports corresponded to about 70% of the officially notified patient load in these two districts managed by the NTP. Even though these districts are not representative of the average district in Viet Nam, they present a compelling argument to expand novel private provider engagement models in the country’s urban areas.

Meanwhile, the efficiency of this approach was evidenced by the high ratio of positively detected cases among those successfully referred. This high ratio suggests a pre-screening step performed by these healthcare professionals or self-selection by patients. The high ratio consequently implies the risk of false-negative assessments and missed opportunities to engage persons with TB. Therefore, more advocacy for providers and the general population to raise top-of-mind awareness about TB is warranted.

As observed on our study and documented by PPM projects in other settings, a referral strategy in isolation remains limited in both novelty and impact [37]. A more comprehensive engagement strategy is required to identify TB patients accessing treatment via the private sector. Including the reported private TB treatments into the NTP’s routine surveillance would have represented a substantial increase in case notifications in the two study districts. However, since these providers did not complete the NTP’s registration process as an accredited PPM partner, the private TB treatment records were not recognized as official notifications. The registration process is arduous and accompanied by external inspections and laborious reporting requirements, which can inhibit PPM participation for TB in Viet Nam [16]. This suggests the need for bold policies that promote private provider participation. This need is well-understood and has shown substantial impact in other settings once addressed [38,39].

Notification gains represent only the initial milestone. While all people with TB detected and notified through the referral strategy were bacteriologically confirmed, we observed low levels of bacteriologic confirmation among private-sector TB treatments, as only one-third was substantiated by a positive sputum test. We further observed that clinical diagnoses and follow-up testing for bacteriologically-confirmed patients oftentimes did not follow national treatment guidelines. As this study focused on case detection, treatment outcomes were optional to report and sparse when collected. Private providers did not employ a systematic follow-up process but also did not permit the study to directly engage their customers for household contact investigations due to fears of reputational damages from breaching patient confidentiality. This has also been observed in other settings [35] and represents a crucial opportunity to improve quality of private-sector TB care. This is particularly the case in light of the low attendance rate on the capacity building sessions offered by the study, as they were not mandatory for study participation. Consequently, while the goal of policy reform should be to remove unnecessary bureaucratic barriers to promote private provider participation, this reform should be designed with the long-term goal of improving quality of care among all stakeholders in mind.

Meanwhile, access to Xpert testing constituted a unique selling proposition of the PPIA to these providers, which they could pass on to their clientele. This study was the first to enable commercial access to Xpert testing for non-PPM providers in Viet Nam, so that the consistent message across size and geography of providers was that the ability to offer NAAT to their clients was a critical catalyst for participation. While this dynamic may be a temporary effect until market access is established through registration and formalization of a commercial distribution channel, intermediary agencies in other settings should leverage these dynamics to build the private provider network. Increased acceptance of Xpert testing has also been observed to result in a reduction of clinical diagnosis [40], so that increasing private-sector Xpert uptake could substantially reduce the rate of over-diagnosis and contribute to improved individual and public health outcomes. Efforts to optimize NAAT access have proven effective in several settings through the Initiative for Promoting Affordable, Quality TB tests [24,38,41].

An important lesson across both strategies was the need to sufficiently power monetary and non-monetary incentives. Evidence suggests that referral and notification incentives can represent a welcome income generation opportunity [42,43]. However, determining the appropriate threshold at which the individual cost-benefit analysis turns favorable is critical. The level of USD 22.07 proved sufficient to elicit private TB treatment reports among some, but it is safe to say that the 14 reporting providers in our study did not constitute the entire spectrum of private TB treatment. For example, risk-averse providers and those with a small caseload may have found the incentive to be insufficient to offset the risk exposure and expected value of penalties of un-notified TB treatment. These incentives may have also created inefficiencies whereby pulmonologists referred persons with TB through our study that would also have been referred in our absence as this level of incentive was high compared to traditionally paid amounts in Viet Nam [19,44,45]. Nevertheless, the costs of incentives paid by our study to detect a person with TB were a fraction of estimated total costs of detecting a new case through other systematic screening strategies [46] and warrant further optimization and evaluation.

A key success factor of the study was the broad coverage and participation of a diverse set of private providers. This was evidenced by the fact that we received referrals from all types of providers listed above and detected TB cases from most provider types. This effectiveness in generating leads and detecting TB patients also suggests that we were able to target the right providers. One reason for this was likely the detailed a priori landscaping and targeting, which allows implementers to have a better sense of the options people have for care seeking and coverage of their interventions [22,47].

Our study faced several limitations. With respect to private-sector TB treatment, our study was observational in nature, so that we did not attempt to change clinical practices. Similarly, we did not systematically incentivize and collect treatment outcomes in this study, but we intend to do so in future engagements. As such, provider willingness to alter behavior to meet international standards of TB care and the extent to which previously mentioned aspiration of improving diagnostic and treatment quality are feasible remain critical research questions to be answered on future studies. Another limitation was that we were only able to verify private TB treatment through reviews and abstractions of data, which were only available in patient records, as private providers did not permit direct engagement of their customers. The study’s implementation area was limited, so that it is necessary to test the model at a greater scale to strengthen the generalizability of these results. Lastly, it also remains unclear, if this model or an adaptation thereof were appropriate in non-urban areas.

Nevertheless, this pilot study has elucidated the potential gains inherent in effective private sector engagement to national and provincial stakeholders in Viet Nam. As has been noted elsewhere, future work should focus on strengthening data systems, including the use of direct electronic data capture to track referrals and loss to follow up between referral and CXR [38]. This work should also employ mechanisms to verify that private TB treatment reports are genuine individuals who have not already been reported elsewhere in the TB notification system. Finally, policy changes are required to facilitate the scale-up of this approach.

## 5. Conclusions

Private providers in HCMC are treating many people with TB who are not reported to the national program, and it is critical to improve engagement approaches that arrive at a system, which allows private providers to notify through the NTP. To achieve public health targets, this system will also need to ensure the highest level of care adherent to national standards. Scaling effective private-sector engagement efforts, such as this enhanced intermediary model, could have a strong impact on the progress towards ending TB, and we recommend the NTP to scale up the model and through it to build capacity for improvements in quality of TB diagnosis and care.

## Figures and Tables

**Figure 1 tropicalmed-05-00143-f001:**
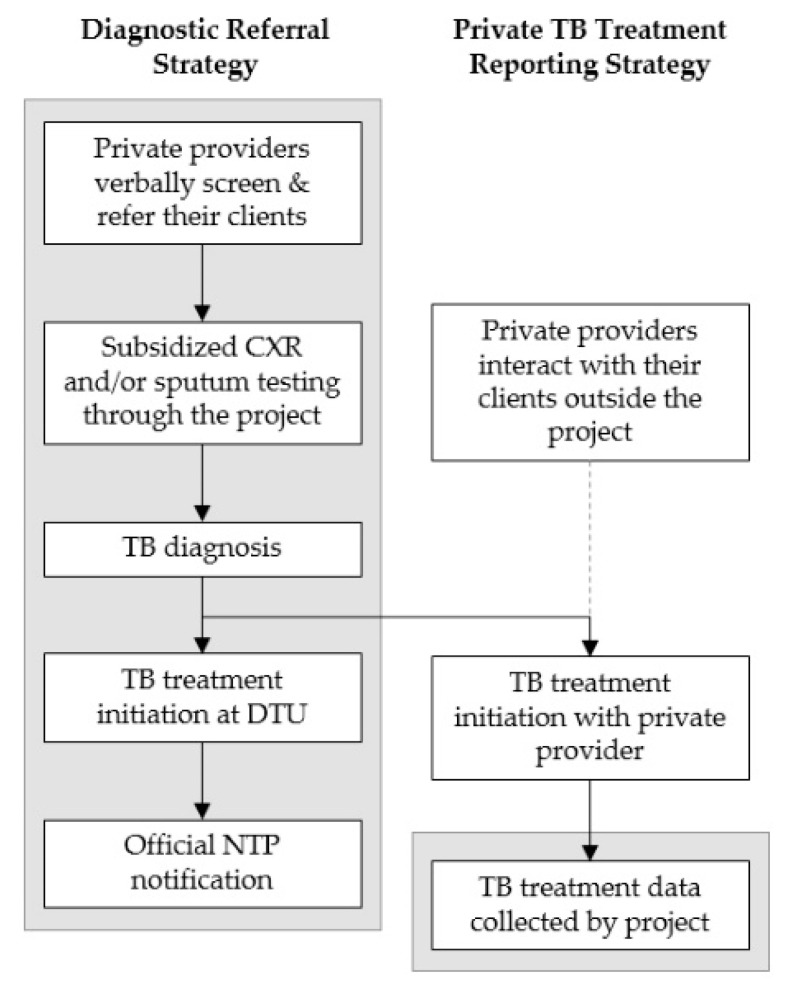
Schematic of the two private sector engagement strategies; the grey boxes show in which parts of the tuberculosis (TB) care cascade private providers were engaged by the study.

**Figure 2 tropicalmed-05-00143-f002:**
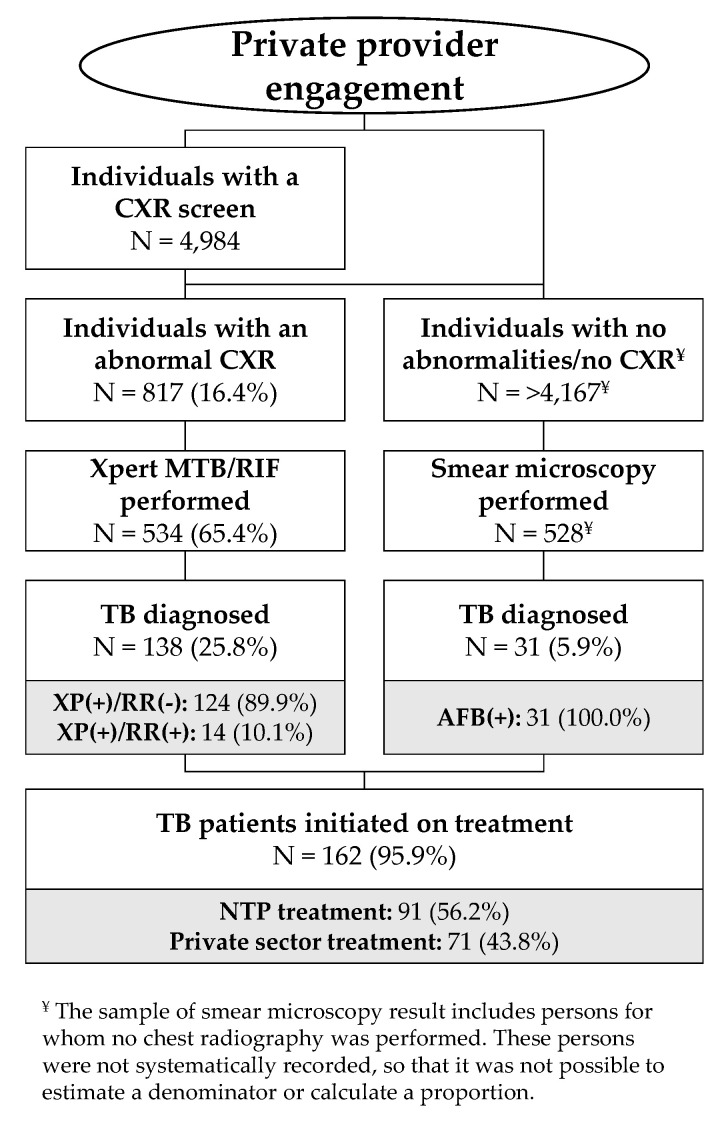
Care cascade among persons screened and referred (2017-Q3 to 2019-Q1).

**Figure 3 tropicalmed-05-00143-f003:**
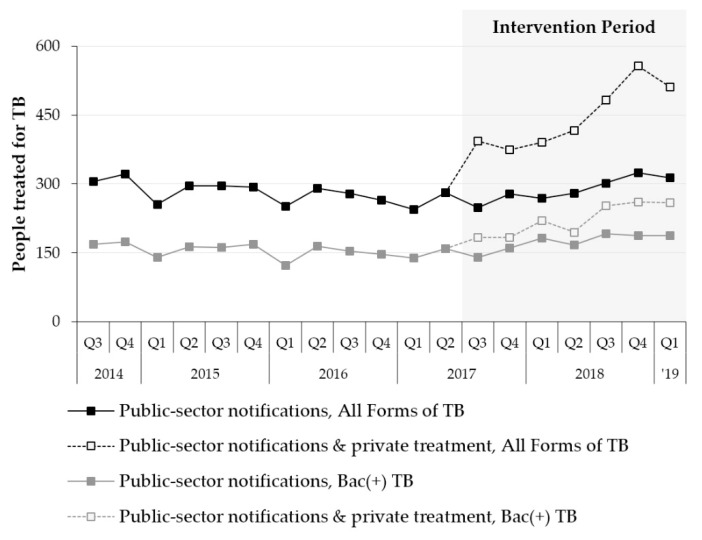
Pre- and post-intervention trends in public-sector TB case notifications and private TB treatment in the study area.

**Table 1 tropicalmed-05-00143-t001:** Summary of provider recruitment and participation referral yields by district (2017-Q3 to 2019-Q1).

	Go Vap	District 10	Total
All licensed private providers	626	481	1107
# deemed eligible for recruitment	469 (74.9%)	273 (56.8%)	742 (67.0%)
# who signed a participation agreement	139 (22.2%)	254 (52.8%)	393 (35.5%)
# trained by provincial lung hospital	119 (19.0%)	72 (15.0%)	191 (17.3%)
# with at least one referral	105 (16.8%)	21 (4.4%)	126 (11.4%)
# reporting private TB treatment	5 (0.8%)	9 (1.9%)	14 (1.3%)

**Table 2 tropicalmed-05-00143-t002:** Summary of chest X-ray (CXR) referrals and Bac(+) TB detection by type of private provider (2017-Q3 to 2019-Q1).

	Providers with Signed Participation Agreement	Providers with 1+ Successful CXR Referral	Successful CXR Referrals	Bac(+) TB Detection
Single doctor clinics	62 (15.8%)	32 (25.4%)	943 (18.9%)	118 (69.8%)
-Pulmonologists	17 (4.3%)	17 (13.5%)	144 (2.9%)	99 (58.6%)
-General practitioners	45 (11.5%)	15 (11.9%)	799 (16.0%)	19 (11.2%)
Multiple doctor clinics	111 (28.2%)	74 (58.7%)	3489 (70.0%)	37 (21.9%)
-Pulmonology specialists	3 (0.8%)	3 (2.4%)	48 (1.0%)	0 (0%)
-Other specialists	108 (27.5%)	71 (56.3%)	3441 (69.0%)	37 (21.9%)
Hospitals	2 (0.5%)	2 (1.6%)	4 (0.1%)	4 (2.4%)
Pharmacies	218 (55.5%)	18 (14.3%)	86 (1.7%)	5 (3.0%)
Community referrals ^1^	N/A	N/A	17 (0.3%)	5 (3.0%)
Undefined provider type	N/A	N/A	445 (8.9%)	0 (0%)
Total	393 (100%)	126 (100%)	4984 (100%)	169 (100%)

^1^ Indicates referrals from a separate community-based ACF initiative that accessed a private sector radiology site for CXR screening.

**Table 3 tropicalmed-05-00143-t003:** Summary characteristics of reported private TB treatment by district.

	Go Vap	District 10	Total
Private providers reporting private TB treatment	5	9	14
Private TB treatment reported	507	605	1112
Average number of privately treated TB patients reported per provider per quarter (range)	14.5 (0–54)	9.6 (0–59)	11.3 (0–59)
Provider type			
Single-doctor practice	263 (51.9%)	389 (64.3%)	652 (58.6%)
Multi-doctor clinic	244 (48.1%)	216 (35.7%)	460 (41.4%)
Diagnosis			
Bacteriologically-confirmed	172 (33.9%)	167 (27.6%)	339 (30.5%)
Clinically diagnosed	335 (66.1%)	438 (72.4%)	773 (69.5%)
Type of TB			
Pulmonary drug susceptible TB	372 (73.4%)	471 (77.9%)	843 (75.7%)
Extra-pulmonary drug susceptible TB	110 (21.7%)	133 (22.0%)	243 (21.9%)
Pulmonary Multi-drug resistant TB	0 (0.0%)	1 (0.2%)	1 (0.1%)
Not reported	25 (4.9%)	0 (0.0%)	25 (2.2%)
Reported residency			
Living in Go Vap or District 10	241 (47.5%)	81 (13.4%)	322 (29.0%)
Living in another district of HCMC	167 (32.9%)	290 (47.9%)	457 (41.1%)
Living outside of HCMC	99 (19.5%)	220 (36.4%)	319 (28.7%)
Not reported	0 (0.0%)	14 (2.3%)	14 (1.3%)
Treatment regimen			
Standard first-line regimen	261 (51.5%)	498 (82.3%)	759 (68.3%)
Modified first-line regimen/no duration	244 (48.1%)	63 (10.4%)	307 (27.6%)
Streptomycin-containing regimen	0 (0.0%)	33 (5.5%)	33 (3.0%)
Levofloxacin-containing regimen ^1^	2 (0.4%)	10 (1.6%)	12 (1.1%)
None reported	0 (0.0%)	1 (0.2%)	1 (0.1%)

^1^ Includes second-line regimen.

**Table 4 tropicalmed-05-00143-t004:** Changes in public-sector TB case notification and private TB treatment by district and type of TB.

	Bac(+) TB	All Forms TB
Go Vap		
Baseline period public-sector TB notifications	703	1315
Intervention period public-sector TB notifications	885	1493
Additional public-sector TB notifications	+182 (+25.9%)	+178 (+13.5%)
Private TB treatment reported during the intervention period	+172 (+24.5%)	507 (+38.6%)
Theoretical additional TB notifications (public & private)	+354 (+50.4%)	+685 (+52.1%)
District 10		
Baseline period public-sector TB notifications	336	544
Intervention period public-sector TB notifications	331	524
Additional public-sector TB notifications	−5 (−1.5%)	−20 (−3.7%)
Private TB treatment reported during the intervention period	+167 (+49.7%)	+605 (+111.2%)
Theoretical additional TB notifications (public & private)	+162 (+48.2%)	+585 (+107.5%)
Both Intervention Districts		
Baseline period public-sector TB notifications	1039	1859
Intervention period public-sector TB notifications	1216	2017
Additional public-sector TB notifications	+177 (+17.0%)	+158 (+8.5%)
Private TB treatment reported during the intervention period	+339 (+32.6%)	+1112 (+59.8%)
Theoretical additional TB notifications (public & private)	+516 (+49.7%)	+1270 (+68.3%)

Baseline period = (2016-Q3 to 2017-Q2)*2 + 2017-Q3. Intervention period = 2017-Q3 to 2019-Q1.

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
