# Peer review of "Enhanced Private Sector Engagement for Tuberculosis Diagnosis and Reporting through an Intermediary Agency in Ho Chi Minh City, Viet Nam"

_tropicalmed, 2020, doi:10.3390/tropicalmed5030143_

Round 1

Reviewer 1 Report

Abstract.

The sentences starting on line 33 and line 35 appear contradicting, because effectively 92.4% of diagnosed patients were not notified to the NTP. So I advise to change in sentence line 35 the text to: Our evaluation showed that an intermediary agency model can potentially engage private providers etc.. 

Line 53. This a general and correct statement that is better placed at the end of the same paragraph. I advise you to add a reference here to a WHO publication on PPM of the period before 2013, e.g. your reference 20. This then demonstrates that NTP has adopted this strategy and policy after a while.

Line 96. Unclear why the two districts have different names, District 10, and Go Vap. Is this an error or an administrative inconsistency? Go Vap district does not have a number? District 10 does not have another name such as Go Vap district has?

Line 132. Not clear what "average ticket per person" means. Can you reformulate that? Does this mean average patient costs per TB treatment at the private practitioner? Also adjust the same expression in line 141.

Question? Why did the project not attempt to incentivize PPs for notification at this stage of the project, because the data of patients were available? The response to this question is given in paragraph 2.4

Line 152. Please adjust text to: Treatment outcomes were not systematically assessed in this pilot. Clarify why "Data provided by providers were sparse because ......". Cohort analysis and treatment outcomes are a key indicator for quality control of TB treatment practices as we know, since WHO launched DOTS in 1993. Did the project not attempt to include this important data collection? Or did the PPs not follow the recording and reporting instructions? This is an important weakness of the project data collection that you need to clarify and account for in the main text and in the Conclusion and Discussion. 

Line 155. I think you should add at the end of that sentence:...and had not undergone the required capacity building and site assessment by NTP.

Line 162. Please rephrase to: Official TB notifications were collected from the two intervention districts for three years prior to the study and during the study period to analyze trends of official TB notifications before and during the pilot.

Line 182. Among participants etc... 48.6% (191/393) is quite a low attendance for capacity building. This weakness you should address in the Discussion section. 

Line 183. By the end of the study we recorded at least one referral for CXR from 32.1% (126/393. This appears low to me. It is good that you provide an analysis of the participating PPs in the consecutive text and Table 1 and 2. 

Table 1 shows striking differences between the two districts on variable 3, 4 and 5 data. Can you provide an explanation for those differences between the two districts?

Table 2 is not clear and correct.

It would be helpful to include a denominator column in table 2 on the left side which states the total selected and registered for the pilot "single doctor clinics" etc for the pilot.  This gives insight into the % of different registered providers with at least 1 successful CXR referral.

The total of column 1 in stated as 126, but is actually 232, not 126. The total of column 2 is 9,416, not 4,989. The total of column 3 is 324, not 169. The percentages are calculated from an incorrect denominator as now stated in the line Total. Please correct the table.

Line 200. I advise to rephrase as follows. The study received TB diagnosis and treatment data from ...If you only mention "treatment reports" then it makes me think of treatment outcome reports. 

Line 201. Rephrase. The top five providers notifying diagnosis and treatment reported private TB treatment for 81.7% (907/1,112) of patients on private TB treatment. 

Line 204. Please add in the first sentence how many and which n/% was bacteriological confirmed among the 1,203 pax diagnosed with TB.

Line 206. You mention Strategy 1 and 2 in this sentence for the first time in this text. That is confusing. Rephrase Figure 1 and introduce in the legend of both columns Strategy 1 and Strategy 2. 

It is clearer that you write that patients diagnosed with rifampicin resistance were largely referred by PPs to NTP facilities (14/15).

Table 3. What is "Average quarterly TB treatment per provider (range)? Please clarify/rephrase. Or delete because the range from 0-54 and 59 does not make much sense to me.

Table 3. Please remove LTBI treatment (n=4) from the title and the data. This is only confusing interpretation as your title in bold of the first column reads "Active All forms of TB". 

Line 240. Not clear. You write here that All forms of TB would increase by 1,270 TB cases. You write in the abstract that the pilot diagnosed 1,203 patients. 

Line 270-273. Please rephrase. .. This high ratio suggests a too strict pre-screening step performed by the healthcare professional or self-selection by patients. deeming the referral unnecessary. The underlined text is confusing, and I propose you delete that. Please delete line 271.

Line 273 rephrase. replace "asymptomatic" nature of TB with "symptomatic" nature. WHO also does not propose to investigate persons who do not have symptoms in the clinical setting. Patients seek care because they have symptoms, or have particular concerns

Author Response

Please see responses under the header "Reviewer 1 comments" in the attachment.

Reviewer 2 Report

The authors submitted a very interesting manuscript which propose strategies to improve TB diagnosis and treatment by engaging private and government healthcare sector in Vietnam. The follow up on patients treatment and care facilities will definitely benefits each individual diagnose and higher recovery frequency.

Author Response

Please see responses under the header "Reviewer 2 comments" in the attachment.

Reviewer 3 Report

The manuscript entitled “Enhanced private sector engagement for tuberculosis diagnosis and reporting through an intermediary agency in Ho Chi Minh City, Viet Nam” by Luan Vo and co-workers is an interesting one.  The authors reported results of their pilot study that deals with an intermediary agency that incentivized private providers in two districts of Ho Chi Minh City to refer persons with presumptive TB and share data of unreported TB treatment from 2017 to 2019. This study highlights the importance of active participation of private providers in Viet Nam to notify people with TB who are not being captured by the current system.    Studies such as reported here will help in Viet Nam’s fight to end TB. The manuscript is well written and appropriately referenced. Thus, I recommend that this manuscript be accepted for publication in Tropical Medicine and Infectious Disease with minor revisions. See some minor comments below.

1. Tuberculosis is one of the top 10 causes of human death worldwide and the leading cause of death from a single infectious agent (ranking above HIV/AIDS and Malaria). According to WHO, in 2018, an estimated 10 million people developed active TB, with 1.2 million succumbing to the disease. Importantly, it is estimated that about one-quarter of the world’s population has latent TB (asymptomatic and noninfectious). So, I recommend the authors provide a brief discussion about TB to give the readers the necessary background about the disease. 

Author Response

Please see responses under the header "Reviewer 3 comments" in the attachment.

Round 2

Reviewer 1 Report

Table 1. The % in the columns are not consistently calculated. The % shown in both columns should be based on the denominator, which is the total shown for both separate districts and for both district together. This is only applied for the first line in each column, 469/626 (74.9%), 273/481 (56.8%), 742/1.107 (67%). This is not applied for the following lines. This needs to be corrected.

Line 215-222 describes numbers of TB patients diagnosed which cannot be seen in any Figure or table. 

Line 223. This line refers to Figure 2. Not 3. Please correct.

It is clear that the PPs diagnose a substantial # of patients with TB for whom there is no bacteriologic confirmation. It would be good to also show this in a Table or Figure, or interpret and discuss this. These diagnoses could include a significant number of patients "over"-diagnosed, if there is no proper quality assurance included in the PPs system. At the same time under-diagnosis may also be a problem of course in the absence of a quality assurance system, that starts with proper training and supervision, either by MOH NTP officials or a dedicated non-government NGO for PPs TB management inspection.

Author Response

31 August 2020

TMID Editorial Office

MDPI

Dear Mr. Matija Petkovic,

We once again sincerely thank you for your consideration of the research article entitled “Enhanced private sector engagement for tuberculosis diagnosis and reporting through an intermediary agency in Ho Chi Minh City, Viet Nam.”

We further thank you for your review and additional comments. Below we have provided a point-by-point response to those additional comments for your review (italics).

Reviewer 1 comments

  • Table 1. The % in the columns are not consistently calculated. The % shown in both columns should be based on the denominator, which is the total shown for both separate districts and for both district together. This is only applied for the first line in each column, 469/626 (74.9%), 273/481 (56.8%), 742/1.107 (67%). This is not applied for the following lines. This needs to be corrected.
    • We have recalculated the percentage as requested.
  • Line 215-222 describes numbers of TB patients diagnosed which cannot be seen in any Figure or table.
    • The results presented in this section represents a combination of the results from Figure 2 and Table 3. As such, we have inserted references for figures mentioned in the section. For example, we referenced “Figure 2” in line 216 and “Table 3” in line 217 to guide the reader on where to find the numbers mentioned.
    • We have added a sentence that breaks down the bacteriologically-confirmed cases into NTP and private sector notifications to clarify the numbers presented in this section. Specifically, we added from lines 217 to 219 the sentence “All 91 TB patients linked to care with the NTP were bacteriologically confirmed. Among persons treated in the private sector, the proportion with bacteriologic confirmation was 30.5% (339/1,112). Together,...”
    • Regarding the MDR-TB cases, thank you for noting this issue. We have edited Table 3 to clearly identify the one MDR-TB cases notified by the private providers. We have further added references to Figure 2 and Table 3 in lines 223 and 224, respectively.
  • Line 223. This line refers to Figure 2. Not 3. Please correct.
    • Thank you for catching this error. We have changed the reference to “Figure 2” as requested.
  • It is clear that the PPs diagnose a substantial # of patients with TB for whom there is no bacteriologic confirmation. It would be good to also show this in a Table or Figure, or interpret and discuss this. These diagnoses could include a significant number of patients "over"-diagnosed, if there is no proper quality assurance included in the PPs system. At the same time under-diagnosis may also be a problem of course in the absence of a quality assurance system, that starts with proper training and supervision, either by MOH NTP officials or a dedicated non-government NGO for PPs TB management inspection.
    • Thank you for this insightful comment. We fully agree with your conclusion.
    • In the results, please refer to Table 3, shows the high proportion of clinical diagnoses (69.5%) to substantiate this finding.
    • In the discussion, please refer to the following sections, where we make a statement concordant to your observation:
      • Lines 302-306: “While all people with TB detected and notified through the referral strategy were bacteriologically confirmed, we observed low levels of bacteriologic confirmation among private-sector TB treatments, as only one-third was substantiated by a positive sputum test. We further observed that clinical diagnoses and follow-up testing for bacteriologically-confirmed patients oftentimes did not follow national treatment guidelines.”
      • Lines 311-313: This has also been observed in other settings [35] and represents a crucial opportunity to improve quality of private-sector TB care. This is particularly the case in light of the low attendance rate on the capacity building sessions offered by the study, as they were not mandatory for study participation.”
      • Lines 325-326: “…so that increasing private-sector Xpert uptake could substantially reduce the rate of over-diagnosis and contribute to improved individual and public health outcomes.”
      • Lines 353-355: “As such, provider willingness to alter behavior to meet international standards of TB care and the extent to which previously mentioned aspiration of improving diagnostic and treatment quality are feasible remain critical research questions to be answered on future studies.”

Thank you very much once again for your review and assessment of our manuscript and please do not hesitate to contact us, if there are any additional requests.

Sincerely and on behalf of the study team,

Luan Nguyen Quang Vo

Friends for International TB Relief

5th Floor, 68B Nguyen Van Troi, 8, Phu Nhuan,
Ho Chi Minh City, Viet Nam

+84-902908004

[email protected]
